# eCONNECT Parent Group: An Online Attachment-Based Intervention to Reduce Attachment Insecurity, Behavioral Problems, and Emotional Dysregulation in Adolescence

**DOI:** 10.3390/ijerph20043532

**Published:** 2023-02-17

**Authors:** Ilaria Maria Antonietta Benzi, Nicola Carone, Marlene Moretti, Laura Ruglioni, Jacopo Tracchegiani, Lavinia Barone

**Affiliations:** 1Lab on Attachment and Parenting—LAG, Department of Brain and Behavioral Sciences, University of Pavia, 27100 Pavia, Italy; 2Department of Psychology, Simon Fraser University, Burnaby, BC V5A 1S6, Canada; 3I.R.C.C.S. Stella Maris Foundation, 56128 Pisa, Italy

**Keywords:** adolescence, attachment, behavioral problems, attachment-based intervention, CONNECT Parent Group, online psychological intervention

## Abstract

During adolescence, a secure parent–adolescent relationship promotes youths’ adjustment and psychological well-being. In this scenario, several studies have demonstrated the effectiveness of the CONNECT program, a 10-session, attachment-based parenting intervention that helps parents understand and reframe their parent–adolescent interactions, reducing adolescents’ insecure attachment and behavioral problems. Furthermore, recent years have witnessed a significant increase in the implementation of effective online versions of psychological interventions, emphasizing the opportunity for more agile and easier dissemination of evidence-based protocols. Therefore, this study aims to identify changes in adolescents’ attachment insecurity, behavioral problems, and parent–child affect regulation strategies, providing preliminary findings on an online, 10-session, attachment-based parenting intervention (eCONNECT). A total of 24 parents (20 mothers, 4 fathers; M_age_ = 49.33, SD = 5.32) of adolescents (M_age_ = 13.83 years, SD = 1.76, 45.8% girls) were assessed on their adolescents’ attachment insecurity (avoidance and anxiety) and behavioral problems (externalizing and internalizing), and on their affect regulation strategies in the parent–child interaction (adaptive reflection, suppression, and affect dysregulation) at three time points: before intervention (t0), after intervention (t1), and at a 2-month follow-up (t2). Mixed-effects regression models highlighted a reduction in adolescents’ internalizing problems (d = 0.11), externalizing problems (d = 0.29), and attachment avoidance (d = 0.26) after the intervention. Moreover, the reduction in externalizing problems and attachment avoidance remained stable at follow-up. Additionally, our findings highlighted a reduction in parent–child affect dysregulation. Results add preliminary evidence on the implementation suitability of an online attachment-based parenting intervention to change at-risk adolescents’ developmental trajectories by reducing attachment insecurity, behavioral problems, and parent–child affect regulation.

## 1. Introduction

Adolescence is a crucial developmental period. Indeed, changes involving the maturation of the body and brain occur during this stage [1,2]. In addition, adolescents face transformations in identity development, in the quality of interpersonal relationships (in family and among friends), and in their capacity for emotional regulation [3]. For these reasons, adolescence also represents a delicate period of psychological development; according to recent data from the WHO, 14% of youths aged between 10 and 19 years experience mental health conditions. Additionally, a recent study on a large Italian sample highlighted that 20% of adolescents reported mental-health-related symptoms [4]. Unfortunately, many remain undetected and untreated, calling for measures and interventions [5].

Indeed, research has shown that, during this developmental period, adolescents are prone to emerging psychopathology with internalizing and externalizing behaviors [6,7]. Anxious, depressive, and somatic aspects, as well as aggressive and behavioral difficulties, significantly impact the quality of their lives; on the one hand, promoting a variety of psychopathological solutions (i.e., substance use and non-suicidal self-injury) [8,9], and on the other hand, affecting the quality of peer relationships, school performance, and mature personality development overall [10,11,12,13,14].

In this scenario, the literature highlights that the quality of the parent–adolescent dyadic relationship configures a delicate balance that can significantly impact the well-being of girls and boys [15]. For example, on the one hand, the adaptive role of parenting characterized by an “authoritative” style has emerged, where aspects of structure and support prevail, integrating elements of emotional involvement and warmth, as well as aspects of limit-setting and consistency [16]. However, on the other hand, a parenting style characterized by physical punishment, criticizing, and scolding is associated with problematic behavioral outcomes in adolescents [17]. Research has also highlighted the bidirectionality of this association, where adolescents’ behavioral problems impact the quality of parenting, thus further underlining the complexity of the dyadic relationship [18].

Moreover, the literature shows that considering attachment relationships represents a central perspective for understanding the organization of adolescents’ affective and behavioral (mal)adaptive experiences [19].

As attachment bonds lay the foundation for social and emotional development and set the stage for future healthy relationships, they can be assessed on both a behavioral and a representational level [20,21]. Assessment of the behavioral level explores behaviors that help ensure that the basic needs for a safe haven and a secure base are met, while the representational level explores the mental representations that influence how individuals think about themselves, others, and relationships in general, and guide their behavior in future significant relationships [22,23].

Thus, as the rapid changes of this developmental period merge with pre-existing attachment needs, adolescents are challenged to explore new territories of identity and self-experience while maintaining the connection and secure basis of attachment relationships with their parents [20]. Against this backdrop, research has underlined the centrality of dealing with internal operating models in adolescence to understand their functioning, dysregulation, and changing mechanisms [24,25]. Specifically, insecure attachment is expressed along two dimensions, anxious and avoidant. Anxious attachment encompasses hyperactivation strategies in response to an inconsistent caregiver, which include worry, rumination, and a state of hypervigilance. Avoidant attachment, on the other hand, in response to a failed relational-proximity-seeking experience, is characterized by deactivating strategies that include emotional suppression and emotional distancing from relationships [26,27]. In a 2013 meta-analysis, Madigan and colleagues considered 60 studies, including 5236 families, and they underscored significant associations between insecure attachment and internalizing behaviors (d = 0.19, 95% CI [0.09, 0.29]) [28]. Similarly, another meta-analysis of 69 studies, including 5947 subjects, highlighted the association between attachment insecurity and externalizing problems (d = 0.31, 95% CI [0.23, 0.40]) [26]. In addition, the results of meta-analyses are inconsistent, with some studies finding that only disorganized attachment strategies, and not organized, insecure ones, are linked to externalizing behavioral problems [21]. In general, studies on the interplay between attachment quality and behavioral problems in adolescence highlight that insecure attachment may be one risk factor (among many others) for later psychopathology [29,30,31].

In the parent–adolescent dyadic relationship, another crucial aspect is the ability to regulate affective states within interpersonal exchanges [32,33]. Indeed, one of the hallmark characteristics of this developmental stage is heightened emotional reactivity, meaning that adolescents may experience intense emotions more frequently and more intensely than they did previously [34]. This increased emotional reactivity can contribute to emotional dysregulation, particularly in relationships between adolescents and their parents [35,36]. In the context of the parent–adolescent relationship, heightened emotional reactivity can make communication and conflict resolution more challenging; parents may feel frustrated by their adolescent child’s behavior, while adolescents may feel misunderstood or unsupported by their parents. In this scenario, the adaptive ability to regulate emotional states coincides with the use of parental reflective capacity; a greater ability to reflect on one’s internal states is associated with a greater ability to regulate intense emotional states in the parent–adolescent relationship [37,38]. On the other hand, suppression, i.e., the denial of emotional states, and affective dysregulation, that is, the very inability to regulate emotional experiences, are maladaptive strategies that foster the worsening of the dyadic interaction [39,40]. Overall, in addition to the ability to influence adolescents’ regulatory processes, parents who can handle their own emotions adequately are better able to take care of their adolescents’ needs by promoting caring behaviors and helping their children not to be overwhelmed by their emotional states.

In this context, implementing attachment-based interventions is crucial to promoting adaptive parent–adolescent interaction and highlight meaningful variables that can impact adolescents’ well-being and mental health [38,41,42].

In recent years, several studies have demonstrated the effectiveness of the CONNECT program, a 10-session attachment-based parenting intervention that helps parents understand and reframe their parent–adolescent interactions, reducing adolescents’ insecure attachment and behavioral problems. The intervention promotes parental sensitivity and reflective capacity in the relationship with adolescent children by recognizing their attachment needs and developmental challenges. Indeed, CONNECT aims to modify insecure attachment at the representational level through the reflective work required in the intervention. At the same time, CONNECT aims at a behavioral-level modification centered more on the construct of parental sensitivity through role-playing and experiential participation in the program. More specifically, the CONNECT intervention focuses on improving parenting behaviors that are related to attachment security; indeed, it promotes sensitive parenting behaviors that are characterized by responsive, supportive, and attuned interactions with adolescents that can help create a secure attachment relationship. Several randomized controlled trials (RCTs) have shown that CONNECT effectively reduces adolescents’ internalizing and externalizing problems and attachment insecurity [43,44,45,46,47]. However, to the best of our knowledge, no contribution provided preliminary data on the CONNECT program about improving dyadic affect regulation strategies.

Moreover, as recent years have witnessed a significant increase in the implementation of effective online versions of psychological interventions [48,49], emphasizing the opportunity for more agile and easier dissemination of evidence-based protocols, the online version of the intervention (eCONNECT) was designed and tested within the context of the pandemic period [50,51]. However, no data are available for this program in regard to the Italian population.

### Aims of the Study

The present study aims to provide preliminary evidence on changes in parental reports of adolescents’ attachment insecurity, behavioral problems, and parent–child affect regulation strategies after completion of an online, 10-session, attachment-based parenting intervention (eCONNECT) attended virtually by a sample of Italian parents of adolescents.

First, based on available research [45,47], we expected parental reports of adolescents’ anxious and avoidant attachment behaviors to decrease over time after the eCONNECT intervention. Second, as suggested by previous contributions [38,43], we expected parental reports of adolescents’ internalizing and externalizing behaviors to decrease after the intervention. Finally, we expected significant improvement in parental dyadic affect regulation strategies after participation in the program.

## 2. Materials and Methods

### 2.1. Study Design and Procedures

The present contribution includes longitudinal data collection. Data are part of the international multicentric study on the eCONNECT program: “Reducing Risk and Promoting Health Among Vulnerable Teens and their Families in the Context of COVID-19”. Data were collected from the University of Pavia and the IRCCS Stella Maris Foundation of Pisa. Parents of adolescents were referred by local mental health centers and schools or were spontaneously approached to receive support for their children’s emotional–behavioral problems. All parents provided informed consent before enrollment. Participation in the study was voluntary, and no incentive was given.

Participants were enrolled in the eCONNECT Parent Group Program. Parents were assessed before intervention (T0), within 2 weeks after the end of the intervention (T1), and at a 2-month follow-up (T2). Conducting a follow-up after a short period of time is in line with previous studies from our research group. Participants received a unique reference code to ensure their anonymity. The Ethical Committees of the Department of Brain and Behavioral Science of the University of Pavia-IUSS and the IRCSS Stella Maris Foundation of Pisa approved all procedures and materials. The research was funded by the Canadian Institutes of Health Research (CIHR) (Grant #448851).

### 2.2. Participants

The sample included N = 24 parents (51% females, M_age_ = 49.33, SD = 5.32) of adolescents (N = 24, M_age_ = 13.83, SD = 1.76). All parents participated in at least 70% of the scheduled sessions. No participants dropped out between T0 and T1, and one dropped out between T1 and T2 (N = 1; 4.2%). The subjects were all Caucasian and fluent speakers of Italian. The eCONNECT groups started in February, April, and September 2021, respectively.

### 2.3. eCONNECT Parent Program Intervention

In each center, two certified leaders delivered the eCONNECT parent group intervention online to 8 participants via 10 weekly, 90-minute sessions. To become certified, each leader had to complete a 3-day specific training, and each received 1 hour of supervision per week on a videotaped intervention.

Each CONNECT session, focusing on adolescence specificities, begins with a discussion of an attachment principle and common challenges in the parent–adolescent relationship (e.g., “conflict is part of attachment”, “autonomy includes connection”, “growth and change are part of relationships”, “conflict is part of attachment”, and “balancing our needs with the needs of others”). In addition, all sessions include experiential role-plays to promote identification and regulation of parents’ emotional reactions to their adolescent’s behavioral problems; encourage reflection on adolescents’ attachment needs; and support adequate responses to adolescents’ behaviors, balancing expectations, and limits. The program explicitly targets four aspects of parenting linked with attachment security in adolescence: caregiver sensitivity, parental reflective function, dyadic affect regulation, and shared partnership/mutuality. The CONNECT intervention’s primary goal is to reinforce the building blocks of secure attachment to reduce parental reliance on unproductive parenting strategies [45]. CONNECT encourages parents to avoid immediate emotional reactions while promoting parental availability, dyadic affect regulation skills, and empathic awareness of the attachment needs underlying their adolescent’s behaviors. Data on the preliminary feasibility of implementing eCONNECT with a Canadian sample are currently available [50].

### 2.4. Measures

#### 2.4.1. Parent-Reported Attachment Insecurity in Adolescents

The Adolescent Attachment Anxiety and Avoidance Inventory—Parent Version (AAAAI-P) [45] is a 36-item, self-report measure of adolescent–parent attachment as rated by parents. The measure includes items from Brennan, Clark, and Shaver’s Experiences in Close Relationships (ECR) scale [52]. Each item is rated on a 7-point scale ranging from 1 = “Strongly Disagree” to 7 = “Strongly Agree”. For this study, we used the reduced 16-item version of the questionnaire. The measure yields an Attachment Anxiety scale (e.g., “My youth wishes that my feelings for him/her were as strong as his/her feelings for me”) and an Attachment Avoidance scale (e.g., “Just when I start to get close to my youth, I find him/her pulling away from me”). The scales showed acceptable to good internal consistency, ranging from α = 0.61 (Attachment Anxiety) to α = 0.85 (Attachment Avoidance). Higher scores on the two scales reflect higher levels of attachment insecurity.

#### 2.4.2. Parent-Reported Behavioral Problems in Adolescents

The Brief Child and Family Phone Interview (BCFPI) [53] is a structured interview administered by telephone or in person to parents to assess emotional and behavioral problems exhibited by 3- to 18-year-olds referred to child mental health services. The revised Ontario Child Health Study scales (OCHS-R) [54] provided the item pool for the BCFPI measures. Each item is scored on a 3-point Likert scale from 0 = “never true” to 2 = “often true”. The BCFPI yields six subscales of emotional and behavioral problems linked to the *DSM* categories of attention-deficit/hyperactivity disorder (ADHD), oppositional defiant disorder (ODD), conduct disorder (CD), seasonal affective disorder (SAD), generalized anxiety disorder (GAD), and major depressive disorder (MDD). The sum of the ADHD, ODD, and CD subscales yields a scale of Externalizing Problems (EXT), and the total score of SAD, GAD, and MDD subscales yields a scale of Internalizing Problems (INT). The scales show good internal consistency: α = 0.85 for EXT, and α = 0.85 for INT. Higher scores on each of the two scales reflect greater behavioral problems.

The Affect Regulation Checklist (ARC) [55] is a 12-item, self-report measure for parents or other caregivers to report on their affect regulation and their child’s affect regulation, as well as a youth self-report version. The ARC comes in different versions to be used depending on the informant and the target; for this study, we used the ARC-R (Relation), which explores the parent–adolescent dyadic affective regulation relationship. Items are rated on a 5-point Likert scale ranging from 1 = “a lot like me” to 5 = “not like me”. The measure yields three scales of Affect Dysregulation (e.g., “I find it very difficult to calm down when I am angry about my child and our relationship”), Affect Suppression (e.g., “I try hard not to think about how I feel about my child and our relationship”), and Adaptive Reflection (e.g., “Thinking about why I feel different emotions for my child helps me learn more about our relationship”). The scales show good internal consistency, ranging from α = 0.72 for Affect Suppression, to α = 0.88 for Affect Dysregulation. Higher scores on each of the scales reflect greater use of the affect regulation strategy.

### 2.5. Statistical Analyses

Statistical analyses were conducted using jamovi version 2.3.13 [56]. General descriptive statistics were computed to describe the sociodemographic characteristics of the participants. To examine changes in attachment insecurity, behavioral problems, and affect regulation strategies in the parent–child interaction, we performed mixed models using the GAMLj package. Using subjects as clusters, time was included as a fixed effect and the intercept for the subject as a random effect to account for within-subject correlations. The mixed-model design allows for the control of the nested nature of the data (i.e., the same participants were assessed across three time points). To control for multiple comparisons, we used the Bonferroni post-hoc test.

## 3. Results

Table 1 shows means and standard deviations for all considered variables at T0, T1, and T2. Table 2, Table 3 and Table 4 show the associations for attachment insecurity, behavioral problems, and affect regulation, respectively, at each time point.

Parents participated in the online program with high satisfaction levels in both centers (Pavia and Pisa), and all parents reported that they were overall satisfied with both the program and the way it was conducted (online), providing encouraging feedback on the online implementation of mental health interventions. In addition, all parents stated they would recommend the eCONNECT program to other families.

### 3.1. Changes in Adolescent’s Attachment Insecurity

Changes in adolescents’ attachment insecurity were explored using two mixed models—one for each outcome (i.e., attachment avoidance and attachment anxiety). First, no change in attachment anxiety was found (F(2) = 1.32, *p* = 0.277, d = 0.13). Second, a significant effect for time was found for attachment avoidance (F(2) = 11.45, *p* = <0.001, d = 0.26). Indeed, parents’ perceptions of adolescent avoidance significantly decreased from T0 and T1 (estimate = −6.21, SE = 1.60, *p* < 0.001); such changes remained stable at T2 (estimate = −7.05, SE = 1.62, *p* < 0.001). Overall, the model explained 75% (R^2^ conditional = 0.75) of the variance in attachment avoidance.

### 3.2. Changes in Adolescent’s Behavioral Problems

Again, changes in adolescents’ behavioral problems were explored using two mixed models—one for each outcome (i.e., internalizing problems and externalizing problems). First, a change in internalizing problems was found (F(2) = 3.87, *p* = 0.028, d = 0.11). Indeed, parents’ perceptions of adolescents’ internalizing problems significantly decreased from T0 to T1 (estimate = −3.13, SE = 1.13, *p* = 0.008). However, changes did not remain stable at T2 (estimate = −1.75, SE = 1.14, *p* = 0.133); internalization scores were higher at T2 (M = 34.88), but they were not significantly different from T0 (M = 36.63) or T1 (M = 33.50). Overall, the model explained 68% (R^2^ conditional = 0.68) of the variance in internalizing problems. Second, a significant effect for time was found for externalizing problems (F(2) = 12.92, *p* < 0.001, d = 0.29). Parents’ perceptions of adolescents’ externalizing problems significantly decreased from T0 to T1 (estimate = −4.79, SE = 1.03, *p* < 0.001), and these changes remained stable at T2 (estimate = −4.24, SE = 1.04, *p* < 0.001). Overall, the model explained 67% (R^2^ conditional = 0.67) of the variance in externalizing problems.

### 3.3. Changes in Dyadic Affect Regulation Strategies

Changes in parent–adolescent dyadic affective regulation strategies were explored using three mixed models–one for each outcome (i.e., dyadic affect dysregulation, affect suppression, and adaptive reflection). First, a change in affect dysregulation was found (F(2) = 7.01, *p* = 0.002, d = 0.18). Parents’ perceptions of dyadic affect dysregulation significantly decreased from T0 to T1 (estimate = −2.38, SE = 0.64, *p* < 0.001), and these changes remained stable at T2 (estimate = −1.50, SE = 0.65, *p* = 0.026). Overall, the model explained 62% (R^2^ conditional = 0.62) of the variance in affect dysregulation. However, no changes were found either in affect suppression (F(2) = 1.65, *p* = 0.204, d = 0.04) or in adaptive reflection (F(2) = 0.91, *p* = 0.410, d = 0.10).

## 4. Discussion

This study aimed to provide preliminary evidence for changes in parental reports of adolescents’ attachment insecurity, behavioral problems, and parent–child affect regulation strategies after an online, 10-session, attachment-based parenting intervention (eCONNECT) in a sample of Italian parents of adolescents.

First, we hypothesized that parental reports of adolescents’ anxious and avoidant attachment would decrease over time after the eCONNECT intervention. However, contrary to the initial hypothesis, there was no significant reduction in parents’ reports of their adolescent children’s anxious attachment even though there was a decrease in reported scores over time. This result could be explained by the online format of the intervention, which may have less effect in containing parents’ anxious attachment issues. On the other hand, a larger sample size and long-term follow-up might contribute to different results, as previous research has demonstrated the effectiveness of the eCONNECT program in reducing attachment insecurity [44].

However, in line with what was initially hypothesized, the findings showed a decrease in parents reporting the avoidant attachment of their adolescents. Thus, eCONNECT promoted a decrease in avoidant strategies characterized by deactivating strategies and affective distancing. Again, this finding is in line with previous studies on the same intervention [43,47].

Second, we hypothesized that parental reports of adolescents’ internalizing and externalizing behaviors would decrease after the intervention. Consistent with the initial hypothesis, the data showed a significant decrease in internalizing and externalizing problems reported by parents after participation in the eCONNECT program. In line with available contributions, the opportunity provided by the intervention for parents to reflect on their relationship with their children and understand their developmental specifics contributed to a decrease in the worsening of adolescents’ behavioral problems [38,45]. However, data showed that the change in internalizing issues was not stable at follow-up. This finding could be related in general to the fact that it is more difficult for parents to detect their children’s anxious and depressive problems as they do not necessarily manifest explicitly as happens with aggressive or dysregulated behaviors. It is possible, again, that a longer follow-up time may help to detect these issues, as well as providing parents with the opportunity to consolidate the strategies learned during the participation in the eCONNECT program [44].

Finally, we hypothesized that we would detect a significant improvement in parental dyadic affect regulation strategies after participation in the program. Interestingly, we only found changes in parent–adolescent dyadic dysregulation (i.e., difficulties in managing and controlling emotions in emotionally charged situations). Indeed, we found no changes, either in the use of the maladaptive strategy of affective suppression (i.e., hiding or masking feelings by controlling facial expressions, body language, and vocal cues to reduce the intensity of negative emotions) or in the ability to reflect on the dyadic relation (i.e., gaining perspective considering the underlying causes of one’s emotions, exploring personal beliefs and attitudes that may contribute to emotional reactivity, and reflecting on how emotions impact behavior and decision-making in the parent–child relationship). No previous study on the CONNECT program has assessed dyadic affective regulation. Indeed, previous studies investigated only adolescents’ dysregulation through parental reports. Dysregulation emerges as a common transdiagnostic mechanism for internalizing and externalizing behavioral problems in this setting [57]. Moreover, at baseline, it emerged as the most frequently used maladaptive strategy. Considering dysregulation within the specific parent–adolescent relationship is a strength of this contribution because it allows us to capture the specificity of parents’ perspectives on their relationships with their adolescents—and not a general perspective on the subjects’ emotional regulation strategies. Indeed, a recent study highlighted significant associations between parent dysregulation and internalizing symptoms (both directly and mediated by attachment anxiety), and between parent dysregulation and externalizing symptoms (both directly and mediated by attachment anxiety and avoidance). Moreover, previous research showed significant associations between parent suppression and internalizing symptoms through attachment anxiety, and between parent suppression and externalizing symptoms through attachment anxiety and avoidance [58]. Thus, these associations pave the way for future studies relying on larger samples.

This study’s results should be understood within the context of the study’s limitations. First, the sample size was small, albeit in a longitudinal data collection. Future studies will benefit from larger sample sizes to account for possible confounding mediators and moderators. Second, we used only parental reports on attachment anxiety and avoidance. Future contributions should also include adolescents’ perspectives. Third, the current findings must be replicated in culturally diverse populations (i.e., data collection in other countries). The multicentric eCONNECT study will provide relevant information on the variables considered in this study. Fourth, effect sizes were small. Although they can still have important practical implications and should not be ignored, further research is needed to (dis)confirm the magnitude of the effects we found. Finally, findings from the present study are exploratory and might not be caused by participation in eCONNECT and its effectiveness; rather, it cannot be excluded that parental reports of the study variables were influenced by parents’ social desirability. In this vein, it is necessary to structure an RCT study to confirm eCONNECT’s efficacy, acknowledging confounding variables and considering differences in treatment-as-usual and untreated samples.

## 5. Conclusions

In conclusion, the results encourage the implementation of the online version of the CONNECT program. Furthermore, the data are promising in suggesting that eCONNECT might foster a reduction of insecure attachment and behavioral problems in adolescents. In addition, the ability of parents to incorporate positive and sensitive parenting promotes adolescents’ well-being and adaptive behaviors during this sensitive developmental stage. In addition, the findings suggest that eCONNECT might contribute to decreasing parent–adolescent dyadic dysregulation; the ability to control affective states in the parent–adolescent relationship is a crucial transdiagnostic factor for dyadic well-being.

All in all, eCONNECT represents a promising tool that offers the possibility of an agile, short-term yet in-depth intervention to provide support to parents of adolescents with behavioral problems who do not have the opportunity to access in-person services.

## Figures and Tables

**Table 1 ijerph-20-03532-t001:** Means and Standard Deviations for Attachment Insecurity, Behavioral Problems, and Emotion Regulation Strategies at T0, T1, and T2.

	T0		T1		T2	
	M	SD	M	SD	M	SD
Attachment Anxiety	26.63	6.51	24.96	6.96	24.30	8.70
Attachment Avoidance	37.42	11.52	31.21	11.50	30.52	9.24
Internalizing Problems	36.63	6.19	33.50	6.78	34.83	7.48
Externalizing Problems	36.13	5.50	31.33	5.94	32.04	5.94
Affect Dysregulation	9.96	3.67	7.58	3.05	8.61	3.60
Affect Suppression	5.75	2.45	6.54	3.30	5.48	2.04
Adaptive Reflection	15.17	3.19	15.96	2.63	15.96	2.95

**Table 2 ijerph-20-03532-t002:** Pearson Correlation Coefficients between Attachment Insecurity at T0, T1, and T2.

	T0 ANX	T1 ANX	T2 ANX	T0 AVO	T1 AVO	T2 AVO
**T0 ANX**	—					
**T1 ANX**	0.54 **	—				
**T2 ANX**	0.54 **	0.59 **	—			
**T0 AVO**	0.07	0.34	0.01	—		
**T1 AVO**	0.11	0.40	−0.11	0.77 ***	—	
**T2 AVO**	0.07	0.15	−0.25	0.67 ***	0.78 ***	—

Note. ANX = Attachment Anxiety; AVO = Attachment Avoidance. ** *p* < 0.01, *** *p* < 0.001.

**Table 3 ijerph-20-03532-t003:** Pearson Correlation Coefficients between Behavioral Problems at T0, T1, and T2.

	T0 INT	T1 INT	T2 INT	T0 EXT	T1 EXT	T2 EXT
**T0 INT**	—					
**T1 INT**	0.62 **	—				
**T2 INT**	0.71 ***	0.70 ***	—			
**T0 EXT**	0.30	0.45 *	0.16	—		
**T1 EXT**	0.20	0.58 **	0.23	0.73 ***	—	
**T2 EXT**	0.32	0.45 *	0.50 *	0.49 *	0.66 ***	—

Note. INT = Internalizing Problems; EXT = Externalizing Problems. * *p* < 0.05, ** *p* < 0.01, *** *p* < 0.001.

**Table 4 ijerph-20-03532-t004:** Pearson Correlation Coefficients between Emotion Regulation Strategies at T0, T1, and T2.

	T0 DYS	T1 DYS	T2 DYS	T0 SOP	T1 SOP	T2 SOP	T0 REF	T1 REF	T2 REF
**T0 DYS**	—								
**T1 DYS**	0.59 **	—							
**T2 DYS**	0.53 **	0.64 ***	—						
**T0 SOP**	0.47 *	0.40	0.26	—					
**T1 SOP**	0.22	0.58 **	0.66 ***	0.35	—				
**T2 SOP**	0.37	−0.00	0.28	0.63 **	0.31	—			
**T0 REF**	−0.17	−0.34	−0.21	−0.20	−0.30	−0.19	—		
**T1 REF**	0.07	0.06	−0.19	−0.32	−0.15	−0.48 *	0.50 *	—	
**T2 REF**	−0.31	−0.07	−0.04	−0.48 *	0.09	−0.46 *	0.28	0.45 *	—

Note. DYS = Affect Dysregulation; SOP = Affect Suppression; REF = Adaptive Reflection. * *p* < 0.05, ** *p* < 0.01, *** *p* < 0.001.

## Data Availability

The datasets generated during and/or analyzed during the current study are available from the corresponding author upon reasonable request.

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
