# Peer review of "eCONNECT Parent Group: An Online Attachment-Based Intervention to Reduce Attachment Insecurity, Behavioral Problems, and Emotional Dysregulation in Adolescence"

_ijerph, 2023, doi:10.3390/ijerph20043532_

Round 1

Reviewer 1 Report

It is a good, well-structured and designed study with clear hypotheses and objectives and a well implemented procedure. However, I think that the wording should be revised at some specific point where it is difficult to understand:

"Interestingly, we found changes only in parent-adolescent dyadic affective dysregulation but in the use of the maladaptive strategy of affective suppression not in the ability to reflect on the dyadic relation." p8 line 304

Additionally, the results appear to contain errors in the headings of some tables:

'Table 4. Means and standard deviations for Attachment Insecurity, Behavioral Problems, and Emotion Regulation Strategies at T0, T1 and T2.'

It also leads me to confusion when significant results are indicated in Pearson's correlation tests, where it states: * p < .05, ** p < .01, *** p < .001. However, the values in the tables marked with two and three asterisks are precisely above these ranges. The values should be reviewed and the cases where the correlation is significant should be confirmed as it may affect the discussion and conclusions.

Reviewer 2 Report

Thank you very much for the opportunity to review this interesting manuscript reporting on the online implementation of the CONNECT program, an attachment-based intervention for parents of adolescents. Along with implementation, adolescent's attachment insecurity, behavioral problems and parent-child affect regulation strategies were measured by parent reports before the program, immediately after the program and at a 2-months follow-up. After the intervention, a reduction in the adolescents' behavioral problems, attachment avoidance as well as in parent-child affect dysregulation was observed. The authors conclude that the online version of the CONNECT program is suitable for implementation.

I really appreciated the approach of adapting an existing, evidence-based program for online use (especially in light of the Covid-19 pandemic contact restrictions and advancing digitalization), and think the presented manuscript makes a valuable contribution here. Although the sample size is quite small (N = 24) and there was no RCT design including a control group the longitudinal study design with three times of measurement should be highlighted. Nevertheless, I have two main concerns and some other notes that prevent me from recommending the paper for publication in its current version.

Main concerns:

A) The first concern is that I think the authors tried to do too much in one study run: The sample size is small and there was no control group, which is understandable when implementing an evidence-based program in an online setting for the first time. However, in this case, a feasibility study would have been useful to check the process, the organization, parent’s acceptance of the online format and to ask for parent’s feedback etc. In the second run and after calculating expected effect sizes and planning sample sizes an evaluation study can be performed.

However, feasibility was only briefly mentioned in the current paper but was not systematically investigated (line 235-239). In addition, the authors mention in the discussion that a multi-centric evaluation study is planned, so the study presented could well have been a feasibility study. Instead, the paper examined outcomes for an effectiveness evaluation and also, for the first time, parent-child affect regulation strategies as an additional outcome. As already mentioned, for an evaluation study with an additional exploratory outcome variable, the sample size is too small and a control group including a RCT design is missing. Thus, the manuscript presented does not describe a full feasibility study, nor an evaluation study.

B) The second concern is that all outcome variables to evaluate the effectiveness of the online parent program were exclusively assessed by parent’s reports (and again without controlling for natural changes without the program). All participating parents rated their children’s behavior problems, attachment insecurity as well as parent-child affect regulation strategies; and all parents participated in the attachment-based program. Thus, after participating in such a program, parents will expect changes in adolescent’s attachment security and problem behaviors. How did you control for/ consider parent’s social desirability? At least, this should be discussed very carefully considering data from different sources like child reports/ interviews, observational data etc.

Further comments:

1) Abstract: To give readers a better overview of the results, please add the effect sizes in the abstract.

Introduction:

2) Overall, the introduction is well written; adolescence is described as a vulnerable period for developing psychopathology highlighting the relevance of high qualitative parent-adolescent relationships. It is also well displayed that attachment-based intervention programs can effectively promote parental sensitivity and thus fostering parent-child/ adolescent attachment. Regarding the theoretical framework of this study, I think two important aspects are still missing in the presented manuscript:

                2a) In the introduction it is not yet explained at what level attachment is captured: In early childhood, attachment is usually assessed on a behavioral level; with development, this moves to the level of attachment representations, which are stored and organized in internal working models. As eCONNECT represents an attachment-based parenting intervention aiming to reduce adolescent’s attachment insecurity, it should be clearly explained, how adolescent’s attachment insecurity will be decreased. Please add a brief definition of attachment with a clear distinction of attachment behavior and attachment representation organized in internal working models. In addition, it should be clearly explained if eCONNECT aims to reduce attachment insecurity on a behavioral or representational level.

                2b) In addition, the mechanisms of action is missing: Attachment-based intervention cannot directly foster children’s/ adolescent’s attachment security. In contrast, attachment security is always addressed indirectly via parenting behavior. Thus, these interventions aim to increase sensitive parenting behaviors which are predictive of more attachment security in children. This should be carefully explained in the introduction.

3) Line 77-84: In this section, the authors first report two meta-analyses showing the associations between insecure attachments and internalizing as well externalizing behavior problems. Second, they concluded that it is important to consider the covariation of both constructs.  Here, the association is overestimated and overemphasized; rather, the studies show that insecure attachment may be one risk factor (of many) for later psychopathology as the effect sizes reported in the meta-analyses tend to be small to moderate.. Please display this association more cautiously.
In addition, meta-analytical findings are mixed as some studies only found disorganized (and not organized insecure attachment strategies) to be related to externalizing problem behavior, e.g., Madigan et al., 2016.

Methods:

4) Can you explain and report in the paper why the follow-up measurement was performed 2 months after the end of the program? Since effects of interventions addressing children’s socio-emotional development and attachment security need time to be effective, and internal working models of attachment tend to be relatively stable over time and only slow (and quite small) changes are to be expected, follow-up measurements are often performed about after 1 year.

5) Please indicate when or in which year the data were collected and add this information to the methods section.

6) I encourage the authors to describe the eCONNECT program in more detail so that the reader does not have to search for additional literature. For example, were sessions performed on a weekly basis?

Results:

7) Table 1 is a bit confusing to me as in Table headings, the authors speak of T0, T1 and T2 but in the Table itself, data from T1, T2 and T3 are reported. Please clarify.

8) The authors conducted several mixed models for analyzing their data; and although statistical analyzes seem to be fine and well performed I wonder about the interpretation. For example, for the outcome variables internalizing problems and dyadic affective regulation strategies, the main effect for time was not significant; yet, changes between individual measurement time points are reported as significant and interpreted as such. I think results should only be reported as significant if the overall model or main effect are also significant, otherwise post-hoc tests are not open to interpretation.

Discussion:

9) In the light of the limitations of the study design (see main concerns A & B) the presented results should be interpreted more carefully, e.g., decreases in adolescent’s behavior problems and avoidant attachment could also have been due to parents' reports influenced by social desirability. In addition, the effect sizes are small; this should also be taken into account in the discussion.

Other comments:

10) The keywords are still missing.

11) Some of the citations in the text are not listed in the references, e.g. Bao & Moretti, In press, Tracchegiani et al., In press (line 113) or Moretti, 2203 (line 196). Please make sure that every citation is also listed in the references.

12) The citation style is inconsequent throughout the paper: Sometimes references are cited using chronological numbers and sometimes by using author names. Please select an option and keep it consistent throughout the text.

References:

Madigan, S., Brumariu, L. E., Villani, V., Atkinson, L., & Lyons-Ruth, K. (2016). Representational and questionnaire measures of attachment: A meta-analysis of relations to child internalizing and externalizing problems. Psychological bulletin, 142(4), 367-399.

Reviewer 3 Report

Dear Authors,

I found the topic of implementing an online attachment-based parenting intervention programme (eCONNECT) as very worthwhile and needed. This study aims to identify changes in adolescents’ attachment insecurity, behavioural problems and parent- child affect regulation strategies after an online, ten-session attachment-based parenting intervention programme.

Although I found this manuscript good, well written and comprehensive, I have a few remarks and suggestions for revising before considering for publication in IJERPH.

The title is too long; I suggest finding a shorter title that captures the essence of the article.

The keywords are missing.

Although the introduction is well organised and informative, the authors could mention more specifically the heightened emotional reactivity in adolescence, as this is one of the important aspects contributing to emotional dysregulation between parent and child.

The authors write about the emotional regulation capacities of the parents as the crucial aspect, and they have also measured dyadic affect regulation in the present study. I wonder if they could enrich the analyses with regression models by using affect dysregulation, affect suppression and adaptive reflection as predictors for changes in insecure attachment and internalizing and externalizing problems. I suggest looking at a new paper by Vernon, J. R. G., & Moretti, M. M. (2022). Parent emotion regulation, mindful parenting, and youth attachment: Direct and indirect associations with internalizing and externalizing problems. Child Psychiatry & Human Development

In the table 1 there is a mistake (probably) in the heading (T1, T2, T3).

The two studies referenced in the sentence in lines 281-2 in the Discussion: “On  the other hand, a larger sample size and long-term follow-up might contribute to different results [35,41]” from my point of view do not support your suggestion, because these studies were conducted in other contexts (other research groups in the first study, other aims in the second study).

Your Reviewer

Round 2

Reviewer 2 Report

I think the authors have been very responsible with the comments and the manuscript has improved significantly. I recommend that the manuscript should be accepted for publication.